# Synthesis of Layered Double Hydroxides with Phosphate Tailings and Its Effect on Flame Retardancy of Epoxy Resin

**DOI:** 10.3390/polym14132516

**Published:** 2022-06-21

**Authors:** Hanjun Wu, Wenjun Zhang, Huali Zhang, Pengjie Gao, Lingzi Jin, Yi Pan, Zhiquan Pan

**Affiliations:** 1Key Laboratory of Novel Biomass-Based Environmental and Energy Materials in Petroleum and Chemical Industry, Wuhan Institute of Technology, School of Chemistry and Environmental Engineering, Wuhan 430074, China; wuhj1204@wit.edu.cn (H.W.); fatchoy407@163.com (W.Z.); jinlingzi1208@163.com (L.J.); wuhj1204@cug.edu.cn (Y.P.); wuhj1204@126.com (Z.P.); 2Key Laboratory for Green Chemical Process of Ministry of Education, Wuhan Institute of Technology, Wuhan 430074, China; 3Hubei Provincial Engineering Research Center of Systematic Water Pollution Control, China University of Geosciences, Wuhan 430074, China; 4Hubei Chuxing Chemical Industry Co., Ltd., Yichang 443311, China; cxgpj007@126.com

**Keywords:** phosphate tailings, layered double hydroxides, epoxy resin, fire safety, flame retardancy

## Abstract

In this work, phosphate tailings (PTs) were used as raw materials for the preparation of Ca-Mg-Al layered double hydroxides (LDHs-1) and Ca-Mg-Al-Fe layered double hydroxides (LDHs-2) by co-precipitation method. The as-prepared samples were characterized by FT-IR, SEM, XRD, and XPS and applied as a flame retardant to improve the fire safety of epoxy resin (EP). The results showed that both LDHs-1 and LDHs-2 exhibited layered structure and high crystallinity. Compared with neat EP, the value of limiting oxygen index (LOI) increased from 25.8 to 29.3 and 29.9 with 8 wt% content of LDHs-1 and LDHs-2, respectively. The flame retardant properties of the composite material were characterized by cone calorimeter (CC), and the results showed that the peak value of the smoke production rate (SPR) decreased more than 45% and 74%, total smoke production (TSP) reduced nearly 64% and 85% with the addition of LDHs-1 and LDHs-2. Meanwhile, the value of the total heat release (THR) reduced more than 28% and 63%. The conversion from LDHs to layered double oxide (LDO) might be conducive to the fire safety of EP. Moreover, the transformation of Fe-OH to Fe-O could promote the early cross-linking of polymer. In summary, LDHs-2 could significantly improve the carbonization process of EP and suppress the smoke released during the combustion process.

## 1. Introduction

Phosphate tailings (PTs) are one of the main by-products of phosphate flotation process, which can be divided into positive tailings and reverse tailings according to different flotation methods [1,2]. Meanwhile, all tailings were constitution with main chemical components (CaO, MgO, Al_2_O_3_, SiO_2_, P_2_O_5_) and microcomponents (K_2_O, Na_2_O, Fe_2_O_3_, etc.) [3,4,5]. In China, nearly two billion tons of tailings are produced annually [6,7]. As well, according to the existing phosphate flotation process, 0.44 tons of PTs are produced per ton of phosphate concentrate produced [8,9]. A large amount of PTs not only required a large amount of land storage and expenditure of management, but also caused serious environmental pollution, such as water contamination, air and soil pollution. Therefore, comprehensive utilization and recycling of PTs are very crucial for the sustainable development of phosphorus chemical industry.

Epoxy resin (EP) is widely applied in automotive, national defense, aerospace, and other industries based on its good mechanical properties and chemical stability. However, EP is limited in its application due to its flammability. Therefore, flame-retardant modification of EP is a feasible method to improve its safety and durability [10,11]. In addition, additives can effectively improve the mechanical properties of polymer materials, especially inorganic nanoparticles [12,13,14]. Currently, flame retardants are roughly divided into organic flame retardants and inorganic flame retardants. Among them, organic flame retardants have good flame retardant properties, but the general halogen, N, and P-containing organic flame retardants are in EP mixed combustion will bring a lot of black smoke, and may also emit a lot of toxic gas, which cause serious harm to environment and mankind health [15,16,17]. Thus, more and more researchers are paying attention to the development of inorganic flame retardants, such as modified carbon nanotubes [18], finely-ground ocher [19], filler-diorite [20], etc. Although the flame retardant performance of inorganic flame retardant is slightly inferior to that of organic flame retardant, it has good smoke suppression performance during combustion, which is beneficial to environmental protection [21,22].

Layered double hydroxides (LDHs) is a typical double metal hydroxide with a layered structure, which is similar to brucite, and its general chemical formula is [M^2+^_1−x_ M^3+^_x_ (OH)_2_]_x_ + (A^n−^)_x/n_•mH_2_O, where M^2+^ and M^3+^ represent the divalent and trivalent metal ions, A^n−^ is the interlayer anion [23,24,25,26]. LDHs have been widely reported as a new functional material and flame retardant because of their special anion exchangeability, tunable pore size, and variability. Mao et al. used dolomite as the raw material to prepare ternary hydrotalcites with good crystallinity through a simple co-precipitation method [27]. Shen et al. found that magnesium hydroxide (MH) blended with ethylene-vinyl acetate copolymer showed significant flame-retardant efficiency even when the dosage of MH was lowered to 20 parts per hundred resins [28]. Yang et al. studies showed that P_3_O_10_^5−^ pillared Mg/Al hydrotalcite doped polypropylene (PP) matrix had gained better flammability and mechanical properties than pure PP matrix [29]. PTs are rich in divalent and trivalent metal elements, and the use of PTs to prepare hydrotalcite will provide the possibility to reduce the amount of phosphate tailings and develop green flame-retardant materials.

Herein, the present work reported an efficient method to prepare LDHs from PTs by co-precipitation method. The thermal stability, flame retardancy, smoke/CO release rates of as-prepared LDHs/EP were all researched to determine the effect of LDHs on the flame retardant properties of EP composites. Meanwhile, the correlation mechanisms appeared in the studies of condensed phases and gaseous phases of EP. This raw material for the synthesis of green inorganic flame retardant with solid waste is expected to realize the reduction and resource utilization of phosphorus tailings, and effectively enhance the combustion safety of epoxy resin.

## 2. Materials and Methods

### 2.1. Materials

PTs were selected from a phosphorus chemical industry located in Yichang, Hubei province. Magnesium chloride hexahydrate (MgCl_2_·6H_2_O), aluminum chloride hexahydrate (AlCl_3_·6H_2_O), iron chloride hexahydrate (FeCl_3_·6H_2_O), sodium hydroxide (NaOH), hydrochloric acid (HCl) and 4,4′-Diamino diphenylmethane (DDM) were purchased from Sinopharm Chemical Reagent Co., Ltd. EP (production name: E-44) was purchased by Nantong Xingchen Synthetic Material Co., Ltd. (Jiangsu, China). Deionized water was used in all experimentations. The typical properties and specifications of EP and DDM were showed in Table 1.

### 2.2. Preparation of EP/LDHs Composites

#### 2.2.1. Preparation of Ca-Mg-Al LDHs and Ca-Mg-Al-Fe LDHs

In this work, Ca-Mg-Al hydrotalcites (LDHs-1) and Ca-Mg-Al-Fe hydrotalcites (LDHs-2) were synthesized by acidolysis of PTs and co-precipitation method. Typically, 100 g of PTs was calcined at 900 °C for 3 h. A total of 50 g of calcined PTs were decentralization on 89.2 mL deionized water, and then added 89.2 mL HCl (*w*/*w*, 35%), strried for 30 min at 60 °C, followed by filtration obtain acid solution. Added NaOH solution (200 g/L) to the above acid solution for adjusting the pH vaule of 5, then filtration to get pure solution. Adjusted the molar ratio of Ca^2+^ to Mg^2+^ in pure solution was 0.5, Ca^2+^ and Mg^2+^ to Al^3+^ was 3, added 0.36 mol/L NaOH to adjusted pH value of 10, and then aged at 90 °C for 18 h, filtered, washed with distilled water, freeze dried, and the LDHs-1 was obtained. The preparation of LDHs-2 was similar to LDHs-1, the difference was to adjusted the molar ratio of Ca^2+^ to Mg^2+^ in pure solution was 1, Fe^3+^ to Al^3+^ was 1, Ca^2+^ and Mg^2+^ to Fe^3+^ and Al^3+^ was 2.

#### 2.2.2. Preparation of EP/LDHs Composites

A total of 2 g of LDHs-1 and LDHs-2 dispersed to 20 g EP respectively, added 6 g DDM and stirred at 90 °C for 10 min. poured the mixture into the template, aged at 100 °C for 2 h and 150 °C for 2 h successively in the fluid bed. The composite materials of LDHs-1/EP and LDHs-2/EP were obtained after cooling to room temperature.

### 2.3. Characterization

The Fourier transform infrared spectroscopy (FT-IR) was performed using Nicolet iS50 (Thermo Scientific, Waltham, MA, USA). X-ray diffraction (XRD) was obtained (PANalytical, Holland) using Cu Kα ray with a scan speed of 2° (2θ) min^−1^. Scanning electron microscopy (SEM) images were investigated by SU 8010 (Hitachi, Tokyo, Japan), and the SEM accelerated voltage was 20 kV. The thermogravimetric analysis (TGA) was measured using TA Q5000 (TA Co., Newcastle, DE, USA) at the heating rate of 10 °C min^−1^ from room temperature to 1000 °C under N_2_ condition. Oxygen index tester (HC-2C, Nanjin, China) followed GB/T2406.2-2009 standards detected the level of burning materials. The degree of graphitization of residual carbon was determined via a Laser confocal Raman Spectrometer (SPEX.1403). The combustion heat release, smoke generation, effective combustion heat and smoke toxicity were test by using Cone calorimeter (FTT0007, West Sussex, UK) followed iso 5660 standards and 35 KW radiation intensity.

## 3. Results and Discussion

### 3.1. Characterization of LDHs-1 and LDHs-2

The XRD patterns of LDHs-1 and LDHs-2 were presented in Figure 1a. The clearly exposing planes of crystals were (003), (006) and (009) except for the obvious diffraction peaks of the LDHs phase, implying the successful preparation of LDHs-1 with typical layered structure [30]. The characteristic (009) diffraction peak of LDHs-2 can be identified at 28.6° [31,32], which generates from the addition of Fe. Compared with the XRD patterns of LDHs-1, the peak signal of LDHs-2 was messy. This may be due to the isomorphic substitution of Fe.

The FT-IR spectra was performed to detect the functional groups between LDHs-1 and LDHs-2 layers. As shown in Figure 1b, the absorption peak located at 3461 cm^−1^ was appeared in both LDHs-1 and LDHs-2, which might be explained by -OH stretching [33]. Compared with the free state stretching vibration peak (3600 cm^−1^) of -OH, this peak drifted to a lower wave number, which could be contribute to the hydrogen bonding between the inter layer H_2_O and the laminate -OH. The absorptions bands of -OH from the interlayer H_2_O of LDHs could be observed at 1630 cm^−1^. The appearance of the peak at 1372 cm^−1^ was explained by asymmetric stretching vibration of C-O in CO_3_^2−^ [33]. The peaks located between 400 cm^−1^ and 450 cm^−1^ were attributed to lattice oxygen vibrations of the cations of Mg^2+^, Ca^2+^, Al^3+^ and Fe^3+^, which were consistent with the infrared characteristic lines of LDHs-1 and LDHs-2, implying that the LDHs-1 and LDHs-2 might be synthesized successful [34].

The chemical state of the compositional elements in LDHs-1 and LDHs-2 were revealed by XPS. As shown in Figure 2, the binding energies of Mg 1s in LDHs-1 and LDHs-2 samples were 1303.7 eV and 1304.1 eV, indicating the existence of the Mg(II) [35]. The single peaks located at 351.6 eV and 351.8 eV assigned to the Ca 2p, means there was only one species of oxygen [36]. The peak points at 74.4 eV and 74.3 eV, attributed to Al 2p of LDHs-1 and LDHs-2, atttibuted to the existence of Al(III) in the form of Al-O [37]. The binding energies of Fe 2p1/2 and Fe 2p3/2 in LDHs-2 were 710.5 eV and 724.9 eV, respectively, represented two kinds of oxygen species in Fe_2_O_3_ [38]. The peak located at 531.65 eV and 531.8 eV assigned to O 1s of LDHs-1 and LDHs-2 could belong to the existence of O-metal [39].

FE-SEM was carried out to obtain the microstructure of the prepared samples, and the images were shown in Figure 3a,b. It can be seen that both LDHs-1 and LDHs-2 showed obvious layered structure. The fewer small flaky particles scattered on the surface and inter layers in Figure 3b, could be caused by incomplete crystallization of LDHs-2. Meanwhile, the obvious smooth surface layered morphology can be seen in Figure 3a,b, indicating that LDHs-1 and LDHs-2 had high crystallinity.

The thermal stability of LDHs-1 and LDHs-2 were investigated by TG-DWG, and shown in Figure 1c. There was a weight loss of LDHs-1 and LDHs-2 when the temperature was less than 200 °C, which were attributed to the removal of adsorbed water [40]. As well, the sharp weight loss of LDHs-1 and LDHs-2 when the temperature increased from 200 °C to 350 °C was explained by the removal of interlayer water. The endothermic section during 350–500 °C of LDHs-1 and LDHs-2, indicating that -OH and CO_3_^2−^ of the hydrotalcite layer had transformed to H_2_O and CO_2_ [41]. In other words, the structure of the hydrotalcite was destroyed, and multiple metal composite oxides were generated.

### 3.2. Evaluation of Flame Retardant Performance with EP/LDHs Composites

#### 3.2.1. Thermal Behavior of EP/LDHs Composites

The thermal stability of LDHs-1/EP and LDHs-2/EP composites were investigated by TG under N_2_, as shown in Figure 4. The results indicated that the addition of LDHs-1 and LDHs-2 slightly reduced the initial weight loss of EP, resulting from the removal of LDHs interlayer H_2_O and adsorption of H_2_O. The carbon yield is a significant indicator to determine the thermal stability of composites. The carbon residue rate raised from 22% for pure EP to 25% for LDHs-1/EP and 28.5% for LDHs-2/EP, indicating that the addition of LDHs-1 and LDHs-2 clearly enhanced the thermal oxidation performance of the EP matrix, meanwhile, the LDHs-2 achieved better performance with higher char yield [42]. In summary, LDHs-1 and LDHs-2 both could enhance the thermal stability of EP composites.

#### 3.2.2. Analysis of Limited Oxygen Index

The limited oxygen index (LOI) is one of the important indexes to evaluate the flame retardant properties of the material. Generally, the higher values of LOI represent the more difficult burning of material. The changed trend of LOI under different LDHs additions is shown in Appendix A. With the increase of LDHs addition from 0 to 8%, the values of LOI increased from 25.8% to 29.3% (LDHs-1/EP) and 29.9% (LDHs-2/EP), which indicated that the composite material was gradually converted to refractory material. This might be contributed to the synergistic effect of LDHs and EP, and result in a decreased of total loading of flame retardant additives and the improvement of flame retardancy [43]. Compared with the reported halogen-free flame retardants [44] and organic flame retardants [45], the LOI value of LDHs-1/EP and LDHs-2/EP was slightly lower. However, LDHs-1 and LDHs-2 exhibited a better flame retardant effect compared with other reported inorganic flame retardants [46,47,48]. This may be caused by the partial substitution of calcium hydroxide for magnesium hydroxide in the hydrotalcite structure.

#### 3.2.3. Analysis of Solid Phase Products of Composites after Combustion

The higher the graphitization degree of the carbon residue layer, the denser and better the heat resistance of carbon residue. Raman spectrum was applied to analyze the carbon yield and determine the carbonaceous quality of LDHs-1/EP and LDHs-2/EP. As shown in Figure 5, the appearance of the peaks at 1360 cm^−1^ and 1600 cm^−1^ may be explained by the vibration of the carbon atoms arriving at the disordered graphite (D band) and graphite carbons (G band). The graphitization degree of the materials can be indirectly reflected by the value of the area ratio of the D band and G band (I_D_/I_G_). Meanwhile, the higher ratio of I_D_/I_G_, the less graphitization degree and stronger thermal stability of carbon residue [42]. As shown in Figure 5a–c, pure EP performs a higher ratio of I_D_/I_G_ (3.46) than that of LDHs-1/EP (2.32) and LDHs-2/EP (2.03). It could be because the EP matrix formed more residual carbon with a high graphitization degree under the promotion of LDHs-1 and LDHs-2. Meanwhile, the ratio of I_D_/I_G_ for LDHs-2/EP was higher than that of LDHs-1/EP, indicating that the catalyzed carbon performance of LDHs-1 was better than LDHs-2 [43].

SEM images of the residual carbon were investigated to estimate the flame-retardancy mechanisms. It can be seen in Figure 6, that the fluffy tiny particles of carbon were scattered on the carbon layer of pure EP, while the carbon layers of LDHs-1/EP and LDHs-2/EP were tighter than that of pure EP. The flat carbon layer can effectively inhibit heat transfer and metabolite decomposition, thereby hindering combustion [49]. Moreover, the results showed that LDHs-1 and LDHs-2 had an obvious smoke suppression effect for EP, which was consistent with the results of cone calorimetry.

#### 3.2.4. Thermal Behavior of Composite Combustion Analysis

Heat release rate (HRR), total heat release (THR), CO release rate, smoke production rate (SPR), and total smoke production (TSP) are key results acquired from cone calorimeter (CC) to evaluate inflaming retarding of materials. The CC was used in this study for exploring the thermal behavior of composite materials and the results are shown in Figure 7a,b. It could be seen that there were no significant changes in heat release rate peak (pHRR) of LDHs-1/EP, while more than 45% abatement in THR. Comparatively, there were a nearly 28% decrease in pHRR and a 63% reduction in THR of LDHs-2/EP. The reduced value of pHRR performance indicated that the addition of LDHs could inhibit the combustion of EP, and the flame retardancy of LDHs-2 was better than that of LDHs-1. The decrease of THR indicated that part of EP was incompletely combusted, which could be attributed to the char-forming process, catalyze effect, and the characteristics of LDHs-1/EP and LDHs-2/EP [50].

CO release is a significant element in assessing the safe combustion of EP combustion, which was attributed to the production from incomplete combustion of oxygen-containing groups [51]. The CO release curves of EP, LDHs-1/EP, and LDHs-2/EP were shown in Figure 7c–e. The CO release rate decreased from 10.47 kg/kg of EP to 6.78 kg/kg of LDHs-1/EP and 1.104 kg/kg of LDHs-2/EP. The decrease in CO production showed the enhancement of LDHs on combustion safety of EP. This was mainly because LDHs-1 and LDHs-2 catalyzed EP to reduce the incomplete combustion of oxygen-containing groups.

Smoke production is another important indicator for evaluating the safety of flame retardants. The smoke formation rate (pSPR) and the total smoke production (TSP) curves were shown in Figure 7f,g. Compared with EP, pSPR, and TSP reduced by nearly 45% and 74% for LDHs-1/EP, and 64% and 85% for LDHs-2/EP, respectively. The obvious reduction of pSPR and TSR indicated that LDHs-1 and LDHs-2 had a significant improvement in the combustion safety of EP, and the improvement of LDHs-2 was better than that of LDHs-1 [51].

#### 3.2.5. Mechanism Analysis

Based on the analysis of thermal stability, solid-phase products and thermal behavior of composite combustion, the flame-retardant mechanisms for LDHs-1 and LDHs-2 may be explained by Appendix A. On the one hand, LDHs-1 and LDHs-2 will decompose during combustion and release CO_2_ and H_2_O, which will reduce the concentration of oxygen free radicals in the combustion area and achieve the effect of dilution and quenching [52]. On the other hand, the presence of LDHs-1 and LDHs-2 inhibited smoke and CO release and promoted the char-forming process based on their catalytic performance. LDHs were converted to corresponding oxide (LDO) during combustion, especially Fe-OH transform to Fe-O, which can catalyze the decomposition of EP and promote the early cross-linking of the molecular chain. The result is to improve the carbonization process of EP and suppress the smoke release during the combustion process [53]. Thus, the smoke depression performance of LDHs-2 was better than that of LDHs-1 [54,55]. The results indicated that LDHs-2 and LDHs-1 can effectively improve the fire safety of EP.

## 4. Conclusions

In summary, LDHs-1 and LDHs-2 were prepared from PTs by co-precipitation method and applied to enhance the flame retardancy of EP. The results indicated that the synthesized LDHs-1 and LDHs-2 displayed obviously layered structure and high crystallinity. The addition of both LDHs-1 and LDHs-2 can significantly improve the LOI and flame retardancy of EP. The CC results indicated that the that the smoke suppression of LDHs-2/EP was better than LDHs-1/EP, which might be caused by the conversion from LDHs to LDO in the combustion process. Moreover, the transformation of Fe-OH to Fe-O could promote the early cross-linking of polymer. The result is to improve the carbonization process of EP and suppress the smoke release during the combustion process. Consequently, this research will provide a possibility to prepare a new type of eco-friendly flame retardant materials by resource utilization of PTs.

## Figures and Tables

**Figure 1 polymers-14-02516-f001:**
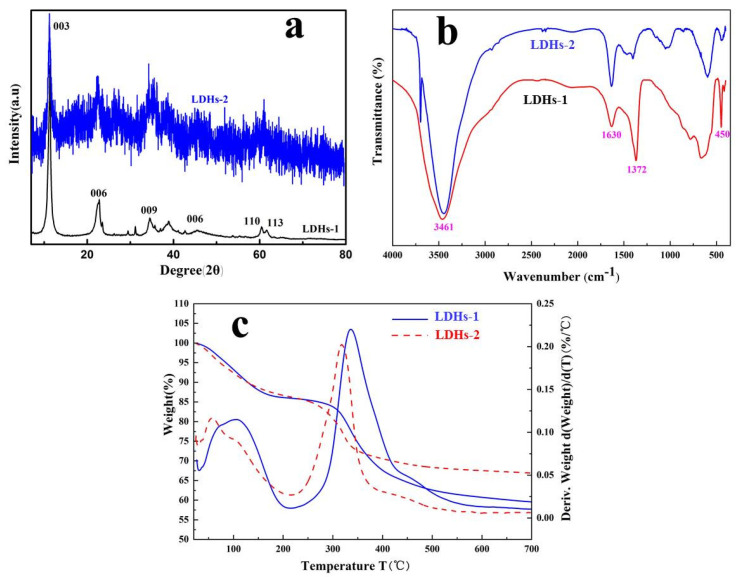
(**a**) XRD curves, (**b**) FT-IR spectras and (**c**) TG-DWG images of LDHs-1 and LDHs-2.

**Figure 2 polymers-14-02516-f002:**
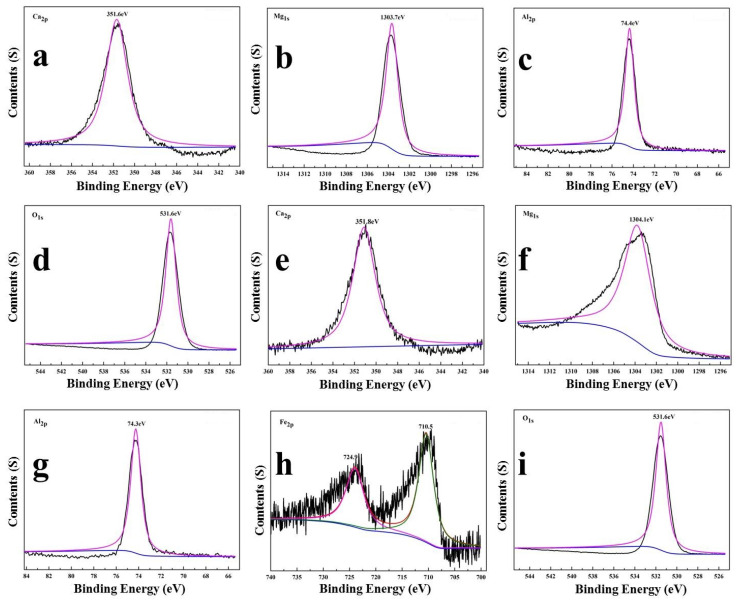
XPS survey (**a**) Ca 2p, (**b**) Mg 1s, (**c**) Al 2p, (**d**) O 1s for LDHs-1 and (**e**) Ca 2p, (**f**) Mg 1s, (**g**) Al 2p, (**h**) Fe 2p, (**i**) O 1s for LDHs-2.

**Figure 3 polymers-14-02516-f003:**
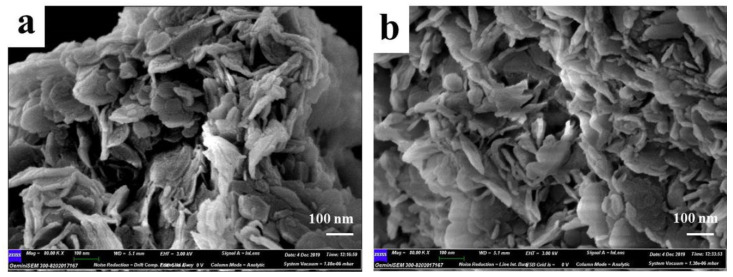
SEM images of (**a**) LDHs-1 and (**b**) LDHs-2.

**Figure 4 polymers-14-02516-f004:**
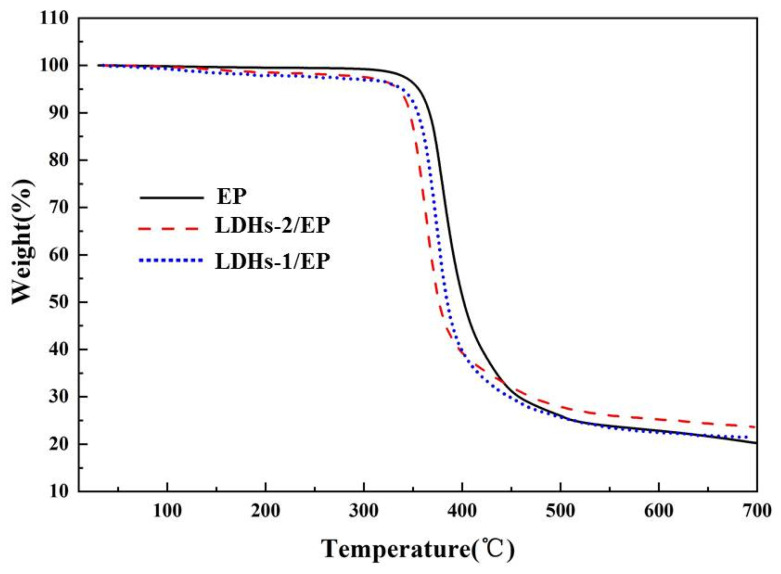
TG images of EP, LDHs-1/EP, and LDHs-2/EP.

**Figure 5 polymers-14-02516-f005:**
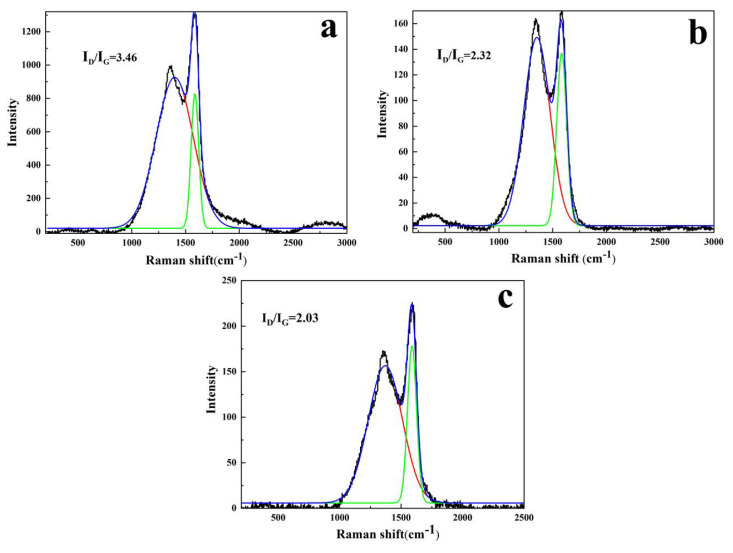
Raman images of (**a**) EP, (**b**) LDHs-1/EP and (**c**) LDHs-2/EP combustion products.

**Figure 6 polymers-14-02516-f006:**
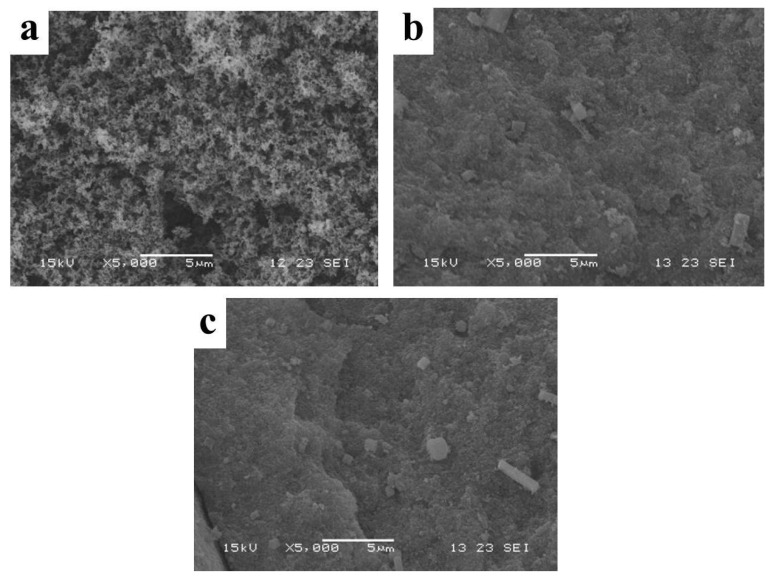
SEM images of (**a**) EP, (**b**) LDHs-1/EP and (**c**) LDHs-2/EP combustion products.

**Figure 7 polymers-14-02516-f007:**
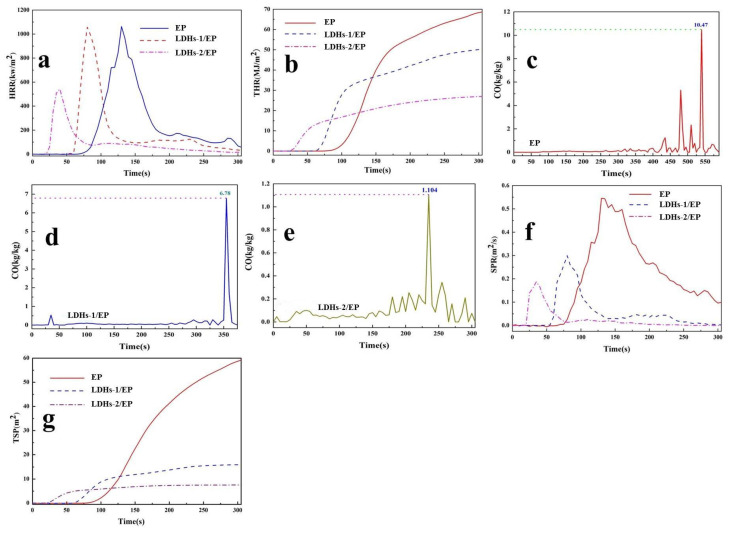
(**a**) HRR, (**b**) THR, (**f**) SPR, (**g**) TSP curves of EP, LDHs-1/EP and LDHs-2/EP; CO release rate curves of (**c**) EP, (**d**) LDHs-1/EP and (**e**) LDHs-2/EP.

**Table 1 polymers-14-02516-t001:** Typical properties and specifications of EP and DDM.

The Qualitative Characteristics	Value
Properties of DDM	
epoxide equivalent, g/mol	168.23
softening point, °C	25
boiling point, °C	264.3
density, g/cm^3^	1.006
Properties of epoxy resin E-44	
epoxide equivalent, g/mol	210–244
epoxide number	0.41–0.47
hydrolysable chlorine, %	≤0.5
inorganic chlorine, mg/kg	≤50
softening point, °C	14–23

## Data Availability

The data presented in this study are available on request from the corresponding author.

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
