# Peer review of "Synthesis of Layered Double Hydroxides with Phosphate Tailings and Its Effect on Flame Retardancy of Epoxy Resin"

_polymers, 2022, doi:10.3390/polym14132516_

Round 1
Reviewer 1 Report
The paper presents an interesting approach based on the Synthesis of layered double hydroxides with phosphate tailings and its effect on flame retardancy of epoxy resin. However, the innovation of the current research work should be further highlighted and emphasized. At the same time, the authors should consider the following comments to greatly improve the quality of the paper.
1. In the list of keywords for this paper, add "flame retardancy".
2. The introduction needs to be improved by relating to the mechanics of the studied materials and their mechanical characteristics. The references to be included are: 10.1007/s10853-022-06994-3, 10.1177/0021998318790093, 10.1016/j.polymertesting.2017.09.009, 10.1177/07316844211051733, 10.1016/j.compstruct.2021.114698 and 10.1016/j.aej.2018.10.012.
3. The text "Preparation of TM LDHs and QM LDHs." needs to be given a sub-heading 2.2.1.
4. The text "Preparation of EP/LDHs composites." needs to be given a sub-heading 2.2.2.
5. The 3 relationships presented in Figure 1 are too small to be read. Kindly enlarge these figures and make sure that the text and numbers included are read-able.
6. The 9 relationships presented in Figure 2 are too small to be read. Kindly enlarge these figures and make sure that the text and numbers included are read-able.
7. For SEM analysis, what was the accelerating voltage applied? what was the working distance? The scale bar is not shown. Kindly either include it or mention the scale value of every 1 cm of the image, as part of the SEM discussion.
8. In the analysis of limiting oxygen index and flame retardancy, kindly relate your results with relevant results from: 10.1177/0731684417727143 and 10.1002/app.46770 and make comparisons in short.
9. The conclusion needs to be modified to summarize the research outcomes in short statements with clear observations.
Author Response
Author’s reply to Editor and Reviews’ comments
We wish to thank the editor and reviewers for the valuable comments on our paper. Our manuscript entitled “Synthesis of layered double hydroxides with phosphate tailings and its effect on flame retardancy of epoxy resin”, was carefully revised according to your comments. The comments and suggestions are incorporated in the revision of the manuscript. The changes in the manuscript have also been marked in red color for your evaluation. The itemized response to each reviewer’s comments is attached. Many thanks for your suggestions.
Reviewer #1:
The paper presents an interesting approach based on the Synthesis of layered double hydroxides with phosphate tailings and its effect on flame retardancy of epoxy resin. However, the innovation of the current research work should be further highlighted and emphasized. At the same time, the authors should consider the following comments to greatly improve the quality of the paper.
- In the list of keywords for this paper, add “flame retardancy”.
Answer: Thanks for your suggestion. Corrections have been made.
- The introduction needs to be improved by relating to the mechanics of the studied materials and their mechanical characteristics. The references to be included are: 10.1007/s10853-022-06994-3, 10.1177/0021998318790093, 10.1016/j.polymertesting.2017.09.009, 10.1177/07316844211051733, 10.1016/j.compstruct.2021.114698 and 10.1016/j.aej.2018.10.012.
Answer: Thanks for your suggestion. Introduction have been improved and reference about the mechanics of the studied materials and their mechanical characteristics was added.
- The text “Preparation of TM LDHs and QM LDHs.” needs to be given a sub-heading 2.2.1.
Answer: Many thanks for pointing this out. Corrections have been made.
- The text “Preparation of EP/LDHs composites.” needs to be given a sub-heading 2.2.2.
Answer: Many thanks for pointing this out. Corrections have been made.
- The 3 relationships presented in Figure 1 are too small to be read. Kindly enlarge these figures and make sure that the text and numbers included are read-able.
Answer: Sorry for our carelessness. Corrections have been made.
- The 9 relationships presented in Figure 2 are too small to be read. Kindly enlarge these figures and make sure that the text and numbers included are read-able.
Answer: Sorry for our carelessness. Corrections have been made.
- For SEM analysis, what was the accelerating voltage applied? what was the working distance? The scale bar is not shown. Kindly either include it or mention the scale value of every 1 cm of the image, as part of the SEM discussion.
Answer: Many thanks for pointing this out. Corrections have been made. The SEM accelerated voltage was 20 kV, and was added in Section 2.3. Figure 3 was modified and added the scale bar.
- In the analysis of limiting oxygen index and flame retardancy, kindly relate your results with relevant results from: 10.1177/0731684417727143 and 10.1002/app.46770 and make comparisons in short.
Answer: This is a good suggestion. I have added a short comparison. “……Compared with the reported halogen-free inorganic flame retardants [44] and organic flame retardants [45], the LOI value of TM LDHs/EP and QM LDHs/EP was slightly lower. This may be caused by the partial substitution of calcium hydroxide for magnesium hydroxide in the hydrotalcite structure”.
- The conclusion needs to be modified to summarize the research outcomes in short statements with clear observations.
Answer: Many thanks for your suggestion. Corrections have been made.
Reference
[44] Zaghloul, M.M.Y.; Zaghloul, M.M.Y. Influence of flame retardant magnesiumhydroxide on the mechanical properties of high density polyethylene composites. J. Reinf. Plast. Compos. 2017, 1802-1816.
[45] Zaghloul, M.M.Y. Mechanical properties of linear low-density polyethylene fire-retarded with melamine polyphosphate. J. Appl. Polym. Sci. 2018, 135, 46770.

Reviewer 2 Report
The title is interesting regarding preparation of flame retardancy properties of epoxy resin by using trimetal layered double hydroxides and quad metal layered double hydroxides. However, I feel that, although the authors have put a big effort to collect information and results, but the novelty is not clear compare to the recent researches.
Additional comments:
1- Improve the abstract based on the results and achievements. Also clarify the pSPR, and THR in abstract.
2- The novelty of the work should be emphasis. Pleas clarify the new idea of the work at the end of introduction.
3- Materials: Please use the full name of the materials.
4- Brief description on figures required. The quality and size of images should be improved. Also the detail of the image is not clear. The y-axis in XRD and TGA should be modified. X-axis in XPS also should be added. A clear scale bar in FESEM should be added. Keep the same for all figures.
5- Please use the journal template for the sections and the format of research paper.
6- As a composite, please investigate the mechanical properties of the different samples.
Author Response
Author’s reply to Editor and Reviews’ comments
We wish to thank the editor and reviewers for the valuable comments on our paper. Our manuscript entitled “Synthesis of layered double hydroxides with phosphate tailings and its effect on flame retardancy of epoxy resin”, was carefully revised according to your comments. The comments and suggestions are incorporated in the revision of the manuscript. The changes in the manuscript have also been marked in red color for your evaluation. The itemized response to each reviewer’s comments is attached. Many thanks for your suggestions.
Reviewer #2:
The title is interesting regarding preparation of flame retardancy properties of epoxy resin by using trimetal layered double hydroxides and quad metal layered double hydroxides. However, I feel that, although the authors have put a big effort to collect information and results, but the novelty is not clear compare to the recent researches. Additional comments:
- Improve the abstract based on the results and achievements. Also clarify the pSPR, and THR in abstract.
Answer: Many thanks for your suggestion. Corrections have been made. The pSPR and THR were clarified in abstract.
- The novelty of the work should be emphasis. Pleas clarify the new idea of the work at the end of introduction.
Answer: Thanks for your suggestion. The idea of this study has been clarified at the end of the introduction. “……This raw material for the synthesis of green inorganic flame retardant with solid waste is expected to realize the reduction and resource utilization of phosphorus tailings, and effectively enhance the combustion safety of epoxy resin”.
- Materials: Please use the full name of the materials.
Answer: Sorry for our carelessness. Corrections have been made. “……Magnesium chloride hexahydrate (MgCl2·6H2O), aluminum chloride hexahydrate (AlCl3·6H2O), iron chloride hexahydrate (FeCl3·6H2O), sodium hydroxide (NaOH), hydrochloric acid (HCl)……”.
- Brief description on figures required. The quality and size of images should be improved. Also the detail of the image is not clear. The y-axis in XRD and TGA should be modified. X-axis in XPS also should be added. A clear scale bar in FESEM should be added. Keep the same for all figures.
Answer: Many thanks for pointing this out. Corrections have been made.
- Please use the journal template for the sections and the format of research paper.
Answer: Thanks for your suggestion. Corrections have been made.
- As a composite, please investigate the mechanical properties of the different samples.
Answer: This is a good question. Indeed, it is necessary to study the mechanical properties of composites. In particular, the effect of the addition of inorganic flame retardant materials on the comprehensive properties of matrix materials. Based on similar inorganic flame retardant materials reported, the mechanical properties of the composites decreased slightly after addition [1]. In this work we focus on the possibility of using solid waste to synthesize green inorganic flame retardants. In the future work, we will study the comprehensive properties (dispersion effect of additives, Mechanical Properties, etc.) of LDHs/EP composites.
Reference
[1] Bekeshev, A.; Mostovoy, A.; Tastanova, L.; Kadykova, Y.; Kalganova, S.; Lopukhova, M. Reinforcement of Epoxy Composites with Application of Finely-ground Ochre and Electrophysical Method of the Composition Modification. Polymers 2020, 12(7), 1437.

Reviewer 3 Report
The manuscript under the title: “Synthesis of layered double hydroxides with phosphate tailings and its effect on flame retardancy of epoxy resin” is in line with Polymers journal. This topic is relevant and will be of interest to the readers of the journal. It based on original research. This research has scientific novelty and practical significance. The article has a typical organization for research articles.
Before the publication it requires significant improvements, especially:
- The "Introduction" section: it has been proven that the effect of various modifying additives and fillers on the flammability reduction and physical and chemical properties of epoxy polymer composites is determined by many factors: ……. I think the related references should be cited corresponding to each aspect, e.g. (but not limited to these), which will undoubtedly improve the "Introduction" section:
- Polymers 2020, 12(7), 1437; https://doi.org/10.3390/polym12071437
- Polymers 2021, 13(19), 3332; https://doi.org/10.3390/polym13193332
· Inorg. Mater. Appl. Res. 2019, 10, 1135–1139, https://doi.org/10.1134/S2075113319050228
· Polymer Composites. 2020; 41: 2025-2035. https://doi.org/10.1002/pc.25517
· Polymers 2021, 13(15), 2421; https://doi.org/10.3390/polym13152421
- The "Introduction" section: show what is the scientific novelty of your research and how it differs from those described in the literature.
- Section 2.1. It is necessary to add the physicochemical characteristics of components - give a table with the main physicochemical and technological properties of epoxy resin and hardener.
- It is necessary to add data on the change in the viscosity of the epoxy composition with the introduction of fillers.
- How was the uniform distribution of fillers in the composition of the epoxy composition achieved? How did you manage to avoid (and did you succeed?) the aggregation of filler particles? It would be nice to confirm this.
- Section 2.3. It is necessary to describe in detail the conditions for conducting thermogravimetric analysis (environment, heating rate, temperature range, etc.).
- Fig.3. The increase in (a) and (b) must be the same, only then can they be compared.
- Please compare achieved results with up-to-date literature, also with composites with other admixtures. Discuss the achieved results.
Author Response
Author’s reply to Editor and Reviews’ comments
We wish to thank the editor and reviewers for the valuable comments on our paper. Our manuscript entitled “Synthesis of layered double hydroxides with phosphate tailings and its effect on flame retardancy of epoxy resin”, was carefully revised according to your comments. The comments and suggestions are incorporated in the revision of the manuscript. The changes in the manuscript have also been marked in red color for your evaluation. The itemized response to each reviewer’s comments is attached. Many thanks for your suggestions.
Reviewer #3:
The manuscript under the title: “Synthesis of layered double hydroxides with phosphate tailings and its effect on flame retardancy of epoxy resin” is in line with Polymers journal. This topic is relevant and will be of interest to the readers of the journal. It based on original research. This research has scientific novelty and practical significance. The article has a typical organization for research articles.
Before the publication it requires significant improvements, especially:
- The “Introduction” section: it has been proven that the effect of various modifying additives and fillers on the flammability reduction and physical and chemical properties of epoxy polymer composites is determined by many factors: I think the related references should be cited corresponding to each aspect, e.g. (but not limited to these), which will undoubtedly improve the “Introduction” section: Polymers 2020, 12(7), 1437; https://doi.org/10.3390/polym12071437
Polymers 2021, 13(19), 3332; https://doi.org/10.3390/polym13193332
Inorg. Mater. Appl. Res. 2019, 10, 1135-1139, https://doi.org/10.1134/S2075113319050228
Polymer Composites. 2020; 41: 2025-2035. https://doi.org/10.1002/pc.25517.
Polymers 2021, 13(15), 2421; https://doi.org/10.3390/polym13152421.
Answer: Thanks for your suggestion. Introduction have been improved and the related references about the mechanics of the studied materials and their mechanical characteristics was added.
- The “Introduction” section: show what is the scientific novelty of your research and how it differs from those described in the literature.
Answer: Thanks for your suggestion. Corrections have been made.
- Section 2.1. It is necessary to add the physicochemical characteristics of components - give a table with the main physicochemical and technological properties of epoxy resin and hardener.
Answer: This is a good suggestion. I have added the Table 1 with the main physicochemical and technological properties of epoxy resin and hardener in Section 2.1.
Table 1. Typical properties and specifications of EP and DDM.
|
The Qualitative Characteristics |
Value |
|
Properties of DDM |
|
|
epoxide equivalent, g/mol |
168.23 |
|
softening point,℃ |
25 |
|
boiling point,℃ |
264.3 |
|
density, g/cm3 |
1.006 |
|
Properties of epoxy resin E-44 |
|
|
epoxide equivalent, g/mol |
210-244 |
|
epoxide number |
0.41-0.47 |
|
hydrolysable chlorine, % |
£0.5 |
|
inorganic chlorine, mg/kg |
£50 |
|
softening point,℃ |
14-23 |
- It is necessary to add data on the change in the viscosity of the epoxy composition with the introduction of fillers.
Answer: This is a good suggestion. The addition of flame retardant materials directly affects the viscosity of EP composites. Further analysis of its impact on the ultimate tensile strength, impact strength and other mechanical properties of EP composites. In this work, we aimed to study the effect of Ca-Mg-Al LDHs and Ca-Mg-Al-Fe LDHs synthesized from solid waste phosphorus tailings on the flame retardant properties of EP, and the effect of hydrotalcite element composition on the flame retardant properties of epoxy resin and combustion smoke release process, while ignoring the change of viscosity and mechanical properties of composite materials after adding flame retardants. In the future work, We will further study the effect of the addition of flame retardants on the viscosity and mechanical properties of EP composites, evaluate their comprehensive properties and the application possibility of this flame retardant material based on phosphorus tailings.
- How was the uniform distribution of fillers in the composition of the epoxy composition achieved? How did you manage to avoid (and did you succeed?) the aggregation of filler particles? It would be nice to confirm this.
Answer: This is a good question. Indeed, the dispersion of inorganic nanoparticles in epoxy resin has always been an intuitive reflection of the superior performance of additives. In this study, we tried to prolong the mixing time of TM LDHs, QM LDHs, DDM molten epoxy resin by stirring at 90 ℃, and then quickly transfer the molten mixture into the mold after stirring. However, this still inevitably leads to the agglomeration of additives. It was necessary to observe and analyze the dispersion of additives in EP by SEM. In this work we focus on the possibility of using solid waste to synthesize green inorganic flame retardants. In the future work, we will study the comprehensive properties (dispersion effect of additives, Mechanical Properties, etc.) of LDHs/EP composites.
- Section 2.3. It is necessary to describe in detail the conditions for conducting thermogravimetric analysis (environment, heating rate, temperature range, etc.).
Answer: Many thanks for pointing this out. Corrections have been made. “……The thermogravimetric analysis(TGA) was measured using TA Q5000 (TA Co., USA) at the heating rate of 10 °C min-1 from room temperature to 1000 °C under N2 condition…..”
- Fig.3. The increase in (a) and (b) must be the same, only then can they be compared.
Answer: Many thanks for pointing this out. Corrections have been made.
- Please compare achieved results with up-to-date literature, also with composites with other admixtures. Discuss the achieved results.
Answer: Thanks for your suggestion. Corrections have been made in Section 3.2.2. “………Compared with the reported halogen-free flame retardants [44] and organic flame retardants [45], the LOI value of TM LDHs/EP and QM LDHs/EP was slightly lower. However, TM LDHs and QM LDHs exhibited better flame retardant effect compared with other reported inorganic flame retardants [46-48]. This may be caused by the partial substitution of calcium hydroxide for magnesium hydroxide in the hydrotalcite structure”.
Reference
[44] Zaghloul, M.M.Y.; Zaghloul, M.M.Y. Influence of flame retardant magnesiumhydroxide on the mechanical properties of high density polyethylene composites. J. Reinf. Plast. Compos. 2017, 1802-1816.
[45] Zaghloul, M.M.Y. Mechanical properties of linear low-density polyethylene fire-retarded with melamine polyphosphate. J. Appl. Polym. Sci. 2018, 135, 46770.
[46] Guo, X.; Wang, H.S.; Ma, D.L.; He, J.N.; Lei, Z.Q. Synthesis of a novel, multifunctional inorganic curing agent and its effect on the flame-retardant and mechanical properties of intrinsically flame retardant epoxy resin. Appl. Polym. Sci. 2018, 29, 46410.
[47] Qian, X.D.; Song, L.; Yu, B.; Wang, B.B.; Yuan, B.H.; Shi, Y.Q.; Hu, Y.; Yuen, R.K.K. Novel organic-inorganic flame retardants containing exfoliated graphene: preparation and their performance on the flame retardancy of epoxy resins. J. Mater. Chem. A 2013, 1, 6822-6830.
[48] Shi, C.L.; Qian, X.D.; Jing, J.Y. Phosphorylated cellulose/Fe(3+)complex: A novel flame retardant for epoxy resins. Polym. Adv. Technol. 2021, 32, 193-189.

Round 2
Reviewer 3 Report
The authors considered most of the comments or adequately responded to the remarks contained in the review; therefore, the work may be approved for publication.